# Chemoradiotherapy but Not Radiotherapy Alone for Larynx Preservation in T3. Considerations from a German Observational Cohort Study

**DOI:** 10.3390/cancers13143435

**Published:** 2021-07-08

**Authors:** Gerhard Dyckhoff, Rolf Warta, Christel Herold-Mende, Volker Winkler, Peter K. Plinkert, Heribert Ramroth

**Affiliations:** 1Department of Otorhinolaryngology, Head and Neck Surgery, University of Heidelberg, 69120 Heidelberg, Germany; rolf.warta@med.uni-heidelberg.de (R.W.); Christel.Herold-Mende@med.uni-heidelberg.de (C.H.-M.); peter.plinkert@med.uni-heidelberg.de (P.K.P.); 2Division of Neurosurgical Research, Department of Neurosurgery, University of Heidelberg, 69120 Heidelberg, Germany; 3Heidelberg Institute of Global Health, University of Heidelberg, 69120 Heidelberg, Germany; volker.winkler@uni-heidelberg.de (V.W.); Heribert.Ramroth@uni-heidelberg.de (H.R.)

**Keywords:** laryngeal cancer, organ preservation, radiotherapy, radiochemotherapy, total laryngectomy, survival

## Abstract

**Simple Summary:**

For advanced laryngeal carcinoma, primary radiotherapy with or without chemotherapy (pCRT or pRT) is used as an alternative to total laryngectomy (TL) to preserve a functional larynx. For advanced laryngeal cancer (T4), poorer survival has been reported after nonsurgical treatment. Is there a need to fear worse survival in moderately advanced tumors (T3)? The outcomes after pRT, pCRT, or surgery were evaluated in 121 patients with T3 laryngeal cancers. pCRT and TL with risk-adopted adjuvant (chemo)radiotherapy (TL ± a(C)RT) yielded results without a significant survival difference. However, after pRT alone, survival was significantly poorer than after TL ± a(C)RT. Thus, according to our data and supported by the literature, pCRT instead of pRT alone is recommended for T3 laryngeal cancers. According to the literature, this recommendation also applies to bulky tumors (6–12 mm), vocal cord fixation, at least minimal cartilage infiltration, and advanced N stage. TL ± a(C)RT instead of larynx preservation should be considered if any of these factors is present and chemotherapy is prohibited; in cases with a tumor volume > 12 mm, severe forms of vocal cord fixation or cartilage infiltration; or when the patient needs a feeding tube or a tracheotomy before the onset of therapy.

**Abstract:**

For advanced laryngeal cancers, after randomized prospective larynx preservation studies, nonsurgical therapy has been applied on a large scale as an alternative to laryngectomy. For T4 laryngeal cancer, poorer survival has been reported after nonsurgical treatment. Is there a need to fear worse survival also in T3 tumors? The outcomes of 121 T3 cancers treated with pCRT, pRT alone, or surgery were evaluated in an observational cohort study in Germany. In a multivariate Cox regression of the T3 subgroup, no survival difference was noted between pCRT and total laryngectomy with risk-adopted adjuvant (chemo)radiotherapy (TL ± a(C)RT) (HR 1.20; 95%-CI: 0.57–2.53; *p* = 0.63). However, survival was significantly worse after pRT alone than after TL ± a(C)RT (HR 4.40; 95%-CI: 1.72–11.28, *p* = 0.002). A literature search shows that in cases of unfavorable prognostic markers (bulky tumors of 6–12 ccm, vocal cord fixation, minimal cartilage infiltration, or N2–3), pCRT instead of pRT is indicated. In cases of pretreatment dysphagia or aspiration requiring a feeding tube or tracheostomy, gross or multiple cartilage infiltration, or tumor volume > 12 ccm, outcomes after pCRT were significantly worse than those after TL. In these cases, and in cases where pCRT is indicated but the patient is not suitable for the addition of chemotherapy, upfront total laryngectomy with stage-appropriate aRT is recommended even in T3 laryngeal cancers.

## 1. Introduction

Laryngeal cancer accounts for most of malignancies in otolaryngology [1]. In 2018, the worldwide incidence rate was 177,422, and 94,771 deaths were caused by laryngeal carcinoma [2]. The male-to-female incidence ratio was 3.6 to 0.5 new cases per 100,000 per year. The age-standardized mortality rates for men and women were 1.9 and 0.3 per 100,000, respectively [3]. The main risk factors for laryngeal cancer are smoking and alcohol consumption [4,5,6].

The paramount aim of tumor treatment is to save the patient’s life but at the same time to preserve his quality of life. This is true especially for laryngeal cancer given the larynx’s crucial role in speaking and swallowing. To this aim, larynx-preservation protocols have been introduced as an alternative to laryngectomy. The benchmark European Organization for Research and Treatment of Cancer (EORTC) and Veterans Affairs (VA) studies have shown that larynx preservation (LP) by radiation after induction chemotherapy is possible “without jeopardizing survival” [7,8]. After demonstrating the superiority of concomitant chemoradiation (CCRT) over induction chemoradiation (ICRT) in terms of LP [9], CCRT has become the standard of therapy in the treatment of advanced laryngeal carcinoma [10]. However, Hoffman, who reviewed almost 160,000 cases of laryngeal squamous cell carcinoma from the National Cancer Database (NCDB), reported decreased survival with an increased use of conservative treatment and a decrease in radical surgery [11]. Particularly in T4 tumors, overall survival after nonsurgical treatment was significantly poorer in many studies [12,13,14,15,16,17,18,19,20]. With respect to T3 laryngeal tumors, the questions regarding whether primary radiotherapy or primary chemoradiotherapy (p[C]RT) results in survival disadvantages or whether these methods can be safely recommended remain open.

## 2. Materials and Methods

The study was approved by the ethics committee of the Medical University of Heidelberg (Ethics Commission S-141/2008 Medical Faculty). As reported previously [21,22,23], all index laryngeal cancer patients were recruited from the five academic tertiary referral centers in the Rhein-Neckar-Odenwald region in southwest Germany between 5 January 1998 and 31 December 2004. Patients in this study either participated in a previous prospective case-control study between 1998 and 2000 or were identified retrospectively through patient records (2001–2004).

Demographic data and clinical information were extracted from hospital medical records and patients were followed up until March 2015. Vital status and date and cause of death for deceased participants were obtained from local registries. Comorbidity conditions were determined using the Charlson Comorbidity Index (CCI) [24]. Details can be found in [22].

Kaplan–Meier (KM) methods and univariable and multivariable Cox proportional hazards models were used to analyze overall survival (OS), disease-specific survival (DSS), and overall survival with functional larynx (OSfL). Multivariable models were adjusted for sex, age, comorbidities, TNM stage at diagnosis, differentiation, and primary tumor site: pCRT and pRT, both with the option of salvage total laryngectomy, were compared with those of surgery with adjuvant radiotherapy or adjuvant chemoradiotherapy as indicated by risk and stage (surgery ± a(C)RT). The proportional hazards assumption was assessed by adding a time-dependent version of all the variables in the model [25]. Details can be found elsewhere [22]. In cases of TL, standardized bilateral neck dissection was performed, in cases of TLM or OPL, neck dissection depended on positive nodal disease. Definitive radiotherapy was performed as conventional 3D-conformal radiation up to 74 Gy, generally in fractions of 2 Gy over 6–7 weeks, in cases of pCRT concurrently with platinum-based chemotherapy, mostly cisplatin 40 mg per square meter of body-surface area weekly. Adjuvant radiotherapy consisted of 50–70 Gy depending on resection status. Data analysis was performed with SAS/STAT software, version 14.2, of the SAS System for Windows, copyright © 2021 (SAS Institute Inc., Cary, NC, USA).

## 3. Results

The demographic and clinical characteristics of the whole study cohort in the different treatment arms are presented in overview in Table 1 and are described in detail elsewhere [22]. The adjuvant treatment of T3 patients is specifically indicated in Table 2.

In stage III, 102 patients received primary surgery, and 16 were treated by definitive conservative therapy. Of these patients, 6 patients received primary radiotherapy (pRT), and 10 received primary chemoradiotherapy (pCRT). After primary surgery, the 5- and 10-year overall survival (OS) rates were 61% and 35%, respectively. After pCRT, the values were 55% and 9%, respectively. After pRT, the values were 14% and 14%, respectively (Table 3). For UICC stage III, in a multivariate Cox regression, OS after pRT was significantly worse than after surgery (hazard ratio (HR) 4.86; 95%-confidence interval (CI): 1.97–12.01; *p* = 0.0006), while after pCRT and after surgery there was no significant difference (HR 1.44; 95%-CI: 0.71-2.94; *p* = 0.31) (Table 4B, Figure 1A).

Among the TL patients, there was a small subgroup of patients who, despite advanced N stage (N2 and N3), did not receive adjuvant therapy (TL−adj; *n* = 4). In univariate analysis, these 4 patients had markedly poorer survival than TL patients who received stage-appropriate adjuvant (chemo)radiotherapy (TL ± a(C)RT) (Figure 1B and Table 4). In a multivariate Cox regression of the subgroup of tumor category T3, no survival difference was noted between pCRT and TL ± a(C)RT (HR 1.20; 95%-CI: 0.57–2.53; *p* = 0.63). However, survival was significantly worse after pRT alone than after TL ± a(C)RT (HR 4.40; 95%-CI: 1.72–11.28, *p* = 0.002) (Table 3). Small numbers of patients treated with pRT and TL−adj must be considered. Low statistical power can be perceived in the large 95%-CI.

DSS, OS, and overall survival with functional larynx (OSfL) after 5 and 10 years of T3 patients in the different treatment groups are given in Table 4.

## 4. Discussion

In our observational cohort study, we reported poorer survival outcomes after pCRT and pRT than after surgery for T4 laryngeal cancer [21]. These findings were blurred by an evaluation over UICC stages III and IV but became evident when evaluating tumor category T4 alone [21]. In the present analysis, for UICC stage III and for tumor category T3, there was no significant difference in survival outcomes after pCRT compared with after surgery (HR 1.44; 95%-CI: 0.71–2.94; *p* = 0.31 and HR 1.20; 95%-CI: 0.57–2.53; *p* = 0.63, respectively, Table 4B,D). However, after pRT alone, there was a significant difference in stage III and in tumor category T3 (HR 4.86; 95%-CI: 1.97–12.01; *p* = 0.0006 and HR 4.4; 95%-CI: 1.72–11.28; *p* = 0.002, respectively, Table 4B,D) (Figure 1A,B).

In the literature, there are conflicting reports regarding whether nonsurgical larynx preservation (LP) is an equivalent therapeutic alternative to surgery for T3 laryngeal cancer. Two different types of larynx preservation studies are noted based on study design: (1) In one type of trial, the aim was to obtain results after nonsurgical LP that were similar to those after surgery, whereas the different efficacies of both strategies were implicitly assumed. (2) In another type of study, the efficacy of both therapeutic approaches was compared.

### 4.1. The Selection of Patients for Different Treatment Modalities Yielded Comparable Results

Nguyen-Tan et al. emphasized the correct selection of patients for the primary conservative approach [26]. For bulky tumors and carcinomas with extensive cartilage invasion, TL combined with aRT was favored. “In appropriately selected patients”, larynx preservation was feasible [26]. Similarly, Timme et al. highlighted the importance of the appropriate choice of T3–T4a patients for primary nonsurgical therapy (*n* = 34) [27]. Factors indicative for primary surgery (total or partial laryngectomy, *n* = 37) included large-volume tumors with distortion of the laryngeal anatomy, fixation of the larynx or the vocal cord, pretreatment higher grade of dysphagia with aspiration, chronic pulmonary disease, and poor performance status caused by multiple comorbidities. The concept of selecting different T3 patients into different treatment arms was reported in a consensus protocol in the Netherlands in 1999 [28]. The primary therapeutic approach for T3 laryngeal cancers was defined as accelerated pRT. In case of N2–3, chemotherapy was supplemented. Operable T4 tumors received laryngectomy, and inoperable T4 tumors underwent pCRT. With this selection, there was no significant difference among surgery, pRCT, and pRT (5-year OS 49%, 45%, and 47%, respectively, *p* = 0.54) [15]. Evaluating the patients between 1999 and 2008 following this concept, Timmermans reported that all T3 patients who deviated from the protocol and initially received laryngectomy were classified as T4 according to the fifth or sixth UICC edition [29]. For evaluation in the study, these patients were reclassified according to the seventh edition, in which the T4 criteria were defined differently. After 2009, for example, infiltration of the cartilage was no longer a T4 criterion but only penetration through the cartilage. Thus, in this study, prognostically favorable T3 patients received pRT, T3N2–3 patients were treated by chemoradiation, and unfavorable T3 patients at the border of T4 patients underwent laryngectomy. This finding explains why outcomes after pRT were superior to pCRT (47% vs. 45%) and why TL yielded comparably poor results (49%). Thus, the data may not be mistaken as results of a prospective randomized trial. These are indeed the results of a sensibly thought-out oncological concept. Applying the accelerated Danish Head and Neck Cancer (DAHANCA) protocol (70 Gy in 35 fractions administered 6 days per week) and choosing only favorable T3 for pRT alone was a successful concept: 5-year OS of 47% after pRT alone compared with 14% after pRT alone in our study. However, in contrast to the Dutch study, we found a negative selection bias for patients who underwent pRT alone in our cohort. Patients in the pRT group had more comorbidities than those in the surgery and pCRT groups (CCI of 0 in only 29% vs. 73% and 76%), and patients were older (median age at primary diagnosis: 64.9 years (pRT) vs. 62.3 years (surgery) and 61.4 years (pCRT)). The question is whether the addition of cisplatin in favorable T3 patients who were treated by pRT alone in the Dutch study would yield even more favorable results.

In T3 laryngeal cancer studies that aim at comparable outcomes between LP and primary surgery, a number of unfavorable criteria were defined that were indicative of adding chemotherapy to radiation or choosing upfront TL instead of LP. In the Dutch concept, N stage N2–3 was an unfavorable criterion indicative of pCRT [15,29]. This notion is consistent with the finding in our observational cohort study: the adjusted HR was 3.5 (95%-CI: 2.2–5.7) for advanced N stage [30]. In many other studies, N stage, especially advanced N stage N2–3, was reported as an independent negative prognostic marker [26,27,31,32].

According to Al-Mamgani et al., in bulky tumors, pCRT should be applied [33]. In supraglottic tumors, thresholds of tumor volumes could be determined, above which oncological outcomes after pRT alone were significantly worse. Consequently, for tumors with volumes greater than 6 or 8 ccm, pCRT instead of pRT should be administered [34,35]. A recent study from Cleveland reported that the locoregional failure rate after pCRT significantly increased for laryngeal cancers >12 ccm regardless of tumor category [36]. Thus, 12 ccm appears to be the upper limit up to which pCRT is sufficiently efficacious. Following the results of these studies, the following limits could be defined for LP in T3. In a patient “unfit” for chemotherapy with a tumor volume of up to 6 ccm, pRT alone would be acceptable. For tumors between 6 and 12 ccm, pCRT appears to be appropriate. For a tumor volume greater than 12 ccm, upfront TL should be considered.

Vocal cord fixation is a further indicator of unfavorable outcomes after pRT alone [37,38]. However, after cisplatin-based pCRT, more favorable results were reported [39]. Vocal cord fixation is not only a negative oncological prognostic factor. It was also described as the strongest predictor of a poor functional outcome after pCRT: 56% of patients with vocal cord fixation persistently needed a feeding tube and/or a tracheostoma for more than 6 months vs. 6% of patients without fixation [40]. As a decision aid, Succo et al. differentiated four patterns of vocal cord fixation: patients with fixation of the arytenoid by weight effect (pattern I) or involvement of the posterior paraglottic space, spreading towards the crico-arytenoid joint, and with infraglottic extension ≤ 10 mm (pattern II) were safely manageable by pCRT. Patients with involvement of the crico-arytenoid joint and infraglottic extension >10 mm (pattern III) and massive crico-arytenoid unit involvement reaching the hypopharyngeal mucosa (pattern IV) should undergo upfront TL [41].

Damage of the perichondrium by tumoral infiltration places the underlying cartilage at considerable risk of perichondritis and subsequent chondronecrosis [42]. Thus, traditionally, cartilage infiltration is considered to be a relative contraindication for radiation therapy [43]. Castelijns noted that cartilage infiltration combined with tumor volumes ≥ 5 ccm significantly worsened the prognosis [44]. According to the sixth edition of the UICC staging system, paraglottic space invasion or minor thyroid cartilage erosion were added to vocal cord fixation as factors for the T3 category [45]. Murakami confirmed paraglottic space invasion as an independent prognostic high-risk factor [46]. Thus, pRT alone was not recommended in T3 with cartilage infiltration [47]. However, according to the consensus panel on larynx preservation clinical trial design, minimal cartilage invasion was regarded to be eligible for pCRT [48]. Forastière et al. emphasized that their finding that pCRT was a suitable approach for achieving locoregional control did not apply to patients with gross destruction of cartilage [9]. In addition, Kamal reported a significantly better 5-year local control rate in patients with non/limited invasion than in those with multiple invasion extension [49]. Following the results of these studies, if a patient was “unfit” for chemotherapy and if there was no cartilage infiltration, pRT alone would be acceptable. For patients with minimal infiltration, pCRT appears to be appropriate. In cases of gross or multiple infiltration of the cartilage, upfront TL should be considered.

Timme described one further important exclusion criterion for LP: pretreatment moderate to severe dysphagia and aspiration [27]. Accordingly, Lefebvre reported laryngeal dysfunction, which was defined as pretreatment tracheotomy, tumor-related dysphagia requiring a feeding tube, or recurring pneumonia within the preceding 12 months requiring hospitalization, as an exclusion criterion for LP clinical trials [50]. After tracheostomy prior to pCRT, Herchenhorn et al. reported a lower rate of complete response (42% vs. 75%; *p* = 0.034) and a significantly poorer 3-year OS (6% vs. 61%, *p* = 0.001). They concluded that previous tracheostomy is such a negative prognostic factor for patients submitted to pCRT that immediate TL instead of LP was recommended [51].

### 4.2. Studies Comparing the Efficacy of Surgical and Nonsurgical Treatment Approaches

Several studies have compared oncological outcomes after primary surgery and primary nonsurgical treatment strategies. The results of these studies depend heavily on the definition of the comparison groups. First, it is important to compare the outcomes of tumor treatments within one tumor category (e.g., T3) not (only) over UICC stages (e.g., stage III and IV). Large randomized prospective larynx preservation studies evaluated all stage III and IV patients together [1,2,3]. As stages III and IV are also defined by the N category, these advanced stages comprised T2, T3, and T4 tumors. Thus, the outcome of a small number of unfavorable tumors could be blurred by the large number of more favorable tumors [21].

Laryngeal and hypopharyngeal cancer can entail laryngectomy. Consequently, in several larynx preservation studies, laryngeal and hypopharyngeal cancers have been evaluated together. However, the biology and prognosis of hypopharyngeal carcinomas are much worse than those of tumors arising from the larynx. Thus, to generate robust data regarding the efficacy of different treatment approaches, the study should evaluate tumors from only one primary tumor site. In the matched-pair analysis of Rades et al., the outcomes after pCRT and after surgery plus a(C)RT of T3 and T4 laryngeal and hypopharyngeal cancers “appeared similar” [52]. This general conclusion may not be applicable to T3 laryngeal cancer as a particular tumor type and category.

Furthermore, to compare the efficacy of different treatment strategies, it makes sense to compare the most efficacious and generally applicable method of treatment in each treatment arm. For pRT alone, as published by Timmermans, accelerated pRT according to the DAHANCA protocol could be suggested as a standard [29]. For pCRT, there may be different opinions as Bonomi notes. In the past, concurrent regimes have been preferred in the US, whereas induction chemotherapy concepts have been favored in Europe [53]. Specific for laryngeal cancer is the larynx preservation study of Forastière et al., which compared CCRT and ICRT head-to-head. Interestingly, while there was only a slight difference after 5 years (58.1% vs. 55.15), long-term OS showed a strong tendency for a superiority of ICCR over CCRT (10-year OS: 38.8% vs. 27.5%; HR 1.25; 95%-CI: 0.98 to 1.61; *p* = 0.08). Licitra et al. pointed out that the long-term results of the Forastière study did not support the superiority of CCRT over ICCR, as the proportion of patients alive with and without larynx was always higher after ICCR [54]. For surgery, laser surgery is an option only in selected cases of T3 laryngeal cancer [55,56]. If it is chosen, the result is largely dependent on the expertise of the surgeon and the caseload of the facility [31]. The most effective and comparable surgery for the comparison of efficacy between surgical and nonsurgical treatment in T3 laryngeal cancers is total laryngectomy (TL). In the study by Rades et al., in the surgical group, only 59% of hypopharyngeal and laryngeal T3 and T4 cancers received total laryngectomy, and 26% of patients had an R1 resection status [57]. Thus, the conclusion on the similarity of the surgical and nonsurgical treatment approaches must be viewed with reservation.

In our study, we chose TL patients as the surgical comparison group. However, as a consequence, there may be a selection bias to more aggressive disease in the surgical cohort because less aggressive disease is more likely to get TLM (in our study, n = 31) or OPL (n = 7) vs. a TL. In our cohort study, in a multivariable Cox regression analysis, there was no statistical difference between TLM and OPL compared with TL (HR 1.1; 95%-CI: 0.63–1.91, *p* = 0.74 and HR 0.52; 95%-CI: 0.18–1.50; *p* = 0.22) (data not shown). However, this must not be misinterpreted as a proof of equivalent therapeutic options. It rather suggests that for less advanced disease less radical surgery was sufficient. Thus, the clinical decision of surgical larynx preservation in these selected patients was acceptable. It is important to note that the rate of necessary adjuvant treatment was markedly higher after TLM compared with after TL (51.6%. vs. 36.9%) (Table 2).

In informed and shared decision-making, the issue of appropriate adjuvant treatment should be discussed overtly. In less advanced T3, TLM may appear to be an option. However, in case histologically free margins cannot be achieved, aCRT is necessary. Thus, the patient might end up getting unnecessary trimodal treatment and subsequent lifelong morbidity, whereas he could have been treated with less intensive bimodal non-surgical treatment or total laryngectomy without the necessity of adjuvant treatment. Surgery— and even radical surgery in form of total laryngectomy—without appropriate adjuvant treatment can turn out to be an inferior therapy. In the TL cohort, we observed a subgroup of four T3 patients who underwent TL but did not receive stage-appropriate adjuvant therapy. This was only a very small number of patients, but they may show a principle descriptively. Three of the four patients died (75%) from locoregional tumor recurrence within 2 years (Figure 1 B), resulting in a 5-year OS rate of 25% (Table 4). Only one patient was a long-term survivor. Our data descriptively show that radical surgical therapy of the primary tumor (i.e., TL) must not stop the patient from accepting consequent stage-appropriate adjuvant treatment. The same phenomenon can be observed in two recent studies. In the large database analysis of Patel [32] including almost 5000 T3 laryngeal cancer patients, this effect was so strong that in the T2–T3 group with high nodal burden (N2–3), a significantly better survival was reported after pCRT than after TL (HR 1.25; 95%-CI: 1.04–1.51 *p* = 0.016). However, this phenomenon was reportedly driven by the patients who—despite given indication—did not receive stage-appropriate adjuvant treatment. This effect was more pronounced in patients with partial resection than in those with total laryngectomy (HR 1.61 vs. HR 1.47). Thus, based on this database analysis, one must not conclude that pCRT in T3N2–3 laryngeal cancers was more efficacious than TL, provided that stage-appropriate aRT is added. Similarly, Su et al. reported a better OS after pCRT in T2 patients than after TL (*p* = 0.036), whereas PFS after TL was superior to pCRT in T3 patients (*p* = 0.005). In sum, the authors concluded that there was no significant difference in PFS and OS between LP and TL. As the study included only UICC stage III and IV patients, T2 patients must have had lymph node metastases. However, in the total laryngectomy group comprising 228 stage III and IV patients, 54% did not receive any adjuvant treatment but received total laryngectomy alone. Thus, a missing adjuvant treatment in the T2 patients who underwent TL might explain their significantly worse survival compared with pCRT. Radical surgical treatment of the primary tumor (TL for T2!) is not a substitute for stage-appropriate adjuvant treatment.

In our cohort study, for T3 patients, TL + stage-appropriate a(C)RT showed significantly better survival than pRT (HR 4.40, 95%-CI: 1.72–11.28) (Table 4D, Figure 1B). After pRT alone, the 5-year OS rate was 14%. After TL ± a(C)RT, it was 58%. After pRT alone, no patient survived 10 years. After TL ± a(C)RT, 35% were long-term survivors (Table 4). However, in our study, in the pRT group, the small number of patients had to be considered (*n* = 7), resulting in a wide 95%-CI, which indicates low statistical power. Our finding is supported by large database studies [13,58]. Regarding laryngeal primaries arising from all subsites, Hoffman et al. reported significantly better survival after surgery than after LP (5-year OS: 64% vs. 49%; *p* < 0.05) [11]. Interestingly, in a subgroup analysis of glottic T3N0 patients, no significant difference in survival outcomes was noted between surgery and pCRT (69% vs. 66%; n.s.), but significantly poorer survival was observed after pRT (48% vs. 69%, *p* < 0.05). This finding is consistent with that of our study. During the recruitment period of our study (1998–2004), the main treatment approach was surgery in Germany, whereas the treatment strategy of 96% of laryngeal cancer patients was pRT alone in neighboring Denmark. Lyhne et al. reported the outcomes of all Danish T3 patients treated by the DAHANCA study group between 1971–2011 (*n* = 713) [59]. The 5-year OS after pRT was 39%, and the 5-year OS with preserved larynx was 24% [59]. In the Danish study, there was no surgical control group. In different cohorts, confounding variables are differently distributed. Thus, univariate data may not be directly compared. However, in Germany, the 5-year OS after TL ± a(C)RT was 58% in our study cohort (Table 4).

Population characteristics in the United States and Denmark may also differ. However, the poorer outcomes in Denmark (39%) [59] compared with the above-reported American database study outcomes of 48% after pRT alone [11,58] may have an additional reason. In the 40-year period, in Denmark, many different radiation regimens have been applied, such as the “gentle curative 52 Gy 13 fraction regimen”, which had significantly worse results than the 5 times per week 66–68 Gy standard (HR 2.3) [59]. The most efficacious of these regimens was the accelerated 66–68 Gy, 6 fractions per week regimen (HR 0.7 compared to standard 5×/week), which was later referred to the DAHANCA protocol [59,60]. Using this DAHANCA regimen as the Netherlands´ standard, Timmermans et al. reached 5-year survival rates of 47% and 51% [15,29], which is the same range as noted in the American results (48%).

In contrast to pRT, in our study, after pCRT compared with after TL ± (C)RT, there was no significant difference in survival outcomes (HR 1.20; 95%-CI: 0.57–2.53; *p* = 0.63, Table 4D). However, in large database studies, less pronounced but still significantly poorer survival after pCRT than after TL was described [11,58]. In a recent study, Bates et al. criticized that in previous database studies, RT dose was not considered, patients who did not receive curative dose of therapy were not excluded, or low thresholds for curative-dose RT were selected [20]. In “an apples-to-apples analysis”, they only compared TL patients who received total laryngectomy and 60–80 Gy of aRT and patients who received pCRT with the full curative dose of 70–80 Gy and any type of chemotherapy [20]. In this trial, pRT alone was no longer regarded as a curative option for T3 laryngeal cancer patients worthwhile of examination. The study confirmed our previous finding that in T4, outcomes after pCRT were significantly inferior to those after TL + aRT (T4N0: HR = 1.39; *p* < 0.001, N4N+: HR = 1.22, *p* = 0.001). However, for T3 tumors, no significant survival difference was noted between the surgical and nonsurgical approaches regardless of N stage: T3N0: HR = 0.94; *p* = 0.38 and T3N+: HR = 0.92, *p* = 0.19. Thus, provided that the full curative dose of 70-80 Gy was applied and that chemotherapy was administered, pCRT yielded equivalent results compared with TL + aRT. This study confirms with high statistical power (*n* = 11,237) the finding of our small number study. After the trials of Cooper and Bernier [61,62], it has become a standard in high-risk patients (i.e., R+ and/or ECS+) to add as adjuvant treatment not only aRT alone but aRT + chemotherapy (aCRT). In Bates’ study, it was not explicitly mentioned in which proportion of the patients in the TL + aRT group this principle was followed. Apart from this, Bates consequently realized the concept that we have proposed above. Specifically, for a reliable comparison between different approaches, the most efficacious treatment in each treatment arm has to be applied. However, we must be aware that these most valuable and real data are the results of selected patients fulfilling special but very reasonable inclusion criteria. These are not the results of a prospective randomized trial reporting the outcomes of all patients intended to treat. The Consolidated Standards of Reporting Trials (CONSORT) flow diagram shows that half of the patients had to be excluded in the last step. A total of 11,026 patients received radiotherapy but did not receive chemotherapy or received a dose of less than 70 Gy and thus did not meet the inclusion criteria [20]. In this case, it is unclear in how many patients the application of a higher dose had been intended. The study of Su provides an impression of the underlying daily clinical life reality. Clinical outcomes of 228 TL patients and 138 pCRT patients were compared [63]. According to the study design, all patients who did not complete a full course of the LP approach due to complications or because patients refused or abandoned the treatment were excluded. Thus, 25% (46 of 184 otherwise eligible patients) were excluded because they died from chemotherapy toxicities (*n* = 3), did not persevere with the treatment (*n* = 35), or the tumor did not respond to induction chemotherapy (*n* = 8). Thus, the study of Bates defines the goal that has to be aimed for to reach equivalent efficacy in pCRT and TL + aRT. This may be an important motivation for the patients and may enhance compliance. However, the study of Bates did not exclusively select patients who successfully received the full curative dose. According to current standards, each individual case was discussed with a tumor board. A differentiated assessment was made by a team of experienced medical and radiation oncologists and oncologic surgeons to determine whether the particular patient was suitable for larynx preservation or might better receive upfront laryngectomy. Thus, even in this study, which properly compares the outcomes of surgical and nonsurgical approaches, the selection criteria mentioned above were more or less explicitly considered in each individual case. Thus, from the study of Bates, we can conclude that for properly selected cases with full curative doses, the 5-year OS rates after pCRT and TL + RT were equivalent. With high statistical power, this result confirms the multivariate result of our study. To date, no study has demonstrated equivalence between LP and TL ± (C)RT evaluating indeed in an “apples to apples” attempt: in the same tumor category, with curative dose and addition of chemotherapy, and with the same risk profile in each treatment arm. This could be realized only in a large prospective randomized trial. Meta-analyses, e.g., [64], which include studies that intended to reach comparable results by explicitly selecting favorable patients for the LP arm [15,26,27,29] or that evaluated patients with tumors of different localizations and T categories together [57], cannot demonstrate equivalence.

However, LP and TL ± (C)RT do not have to be equivalent. If, by proper selection of favorable patients, there is a chance to reach equivalent oncological results and simultaneously preserve a functional larynx, the goal is reached. As the likely more efficacious approach, TL ± (C)RT may be reserved only for unfavorable T3 and T4 patients. Thus, equivalence does not need to be proven, but proper selection of patients for LP must be supported. Ultimately, the individual patient has to decide which approach to choose. However, to make a good decision, the patients have to be well informed. Together with healthcare providers, the patient should ideally be able to approximately assess his/her individual risk of failure. To this end, it would be valuable to establish a larynx preservation failure risk score (LPFRS) analogous to the total dysphagia risk score (TDRS) published by Langendijk et al. [65] (see modification for practical use in [22]). This risk score could be calculated by summation of the independent prognostic factors as categories and their respective feature expression as variables (a. tumor volume: <6 ccm, 6 ccm ≤ x ≤ 12 ccm, or >12 ccm; b. cartilage infiltration: non, minimal, or gross/multiple; c. vocal cord fixation: none, Succo pattern I/II or III/IV; d. N stage: N0–1, N2–3, the latter indicative of pCRT; e. pretreatment laryngeal dysfunction with feeding tube, tracheostomy, or recurring pneumonia in the preceding 12 months = indicative for considering TL instead of LP) multiplied by risk points that would be derived from the regression coefficients from a multivariate model. Ultimately, each patient could be assigned to his individual risk group (low risk ≤ 10%, intermediate risk 10–30%, high risk > 30%), which was defined by recursive partitioning analysis [66]. This LPFRS could be a valuable decision-making aid for the counseling of T3 patients. For patients with low risk, in the case that the patient is unfit for cisplatin-based chemotherapy, pRT might be acceptable in contrast to the general recommendation. For patients with intermediate risk, pCRT would be recommended. For patients with high risk, TL should be considered.

In addition to these tumor-related risk factors, patient-related unfavorable prognostic markers, such as low levels of hemoglobin [67,68,69,70], poor performance status [50,71], age > 70 and comorbidities, are noted. Increased age in itself is a negative prognostic marker. In laryngeal cancer patients, we found that age at first diagnosis was the strongest risk factor (HR 1.5; 95%-CI: 1.5–1.7 per additional 10 years) in multivariate analysis [30]. In advanced-stage laryngeal cancer patients, the general negative prognostic value of age and comorbidities has a stronger specific effect on survival with the LP approach. If the addition of chemotherapy to pRT is contraindicated, with pRT alone, patients receive a treatment, which has significantly poorer outcomes in advanced tumor stages according to several studies, e.g., [11,14,33,58,72]. As stated above, the addition of chemotherapy is an important issue for successful LP in T3 laryngeal cancers. In the case of bulky tumors, gross cartilage infiltration, fixation of the vocal cord, and N2–3 status, if the concept of LP is followed, chemotherapy is highly recommended as pRT alone yielded only poor oncological results. In Great Britain, pCRT is recommended for LP in T3 laryngeal cancer according to the current guidelines [73]. In cases in which the addition of chemotherapy was prohibited due to multiple comorbidities, poor renal function, or advanced age, Lin et al. reported significantly poorer 5-year OS after pRT than after pCRT (13% vs. 48%; *p* = 0.001). With comparable age and comorbidities, we observed almost identical outcomes after pRT alone in our study (14%). The British authors recommend informing patients of this difference in expected mortality to help empower their choice for upfront total laryngectomy instead of an attempt at LP with pRT alone [72]. However, in HNSCC, as Machtay reported, age > 70 years is a significant risk factor for experiencing severe late toxicities after pCRT (1.05 per year (*p* = 0.001)) [74]. Additionally, in two large meta-analyses of up to 25,000 patients in 120 randomized studies, Pignon and Bourhis reported a decreasing favorable effect of chemotherapy on survival with increasing age (trend test up to *p* = 0.003) [75,76]. Concretely, Strom showed in a meta-analysis of 369 patients that patients ≥ 70 years after pCRT experienced more hospitalizations (*p* = 0.02) and had an increased risk of death at 3 months following pCRT (OR 5.19 95%-CI: 1.64–16.4; *p* = 0.005) and suffered worse survival over time (HR 2.30; 95%-CI: 1.34–3.93; *p* = 0.002) [77]. Carboplatin, which is less nephrotoxic but myelodepressive, and the targeted therapeutic agent cetuximab appeared to be of no benefit in patients of advanced age [77]. Following Lin´s recommendation [72], elderly, frail, and multimorbid T3 patients who are not suitable for cisplatin-based chemotherapy should instead undergo upfront laryngectomy.

It is important that patient preferences and willingness for an increased risk of treatment failure are included in the process of decision-making. Spontaneous rejection of recommended laryngectomy is natural and normal. Several studies have shown, however, that in the head and neck and other cancer sites, patients highly value survival and are willing to accept added toxicities and loss of quality of life to maximize their chances of survival [78,79]. Moreover, both pCRT and TL affect, albeit differently, the quality of life (QoL) of patients treated for advanced cancer of the larynx [80]. The general QoL scores of patients after both treatment approaches seemed similar [80]. Interestingly, the effect of laryngectomy even on speech evaluation was not significant between patients after TL + aRT and those after pCRT [81]. Thanks to the possibilities of voice rehabilitation, living without a larynx does not mean living without a voice. Comparing optimal outcomes after pCRT and TL + aRT, patients preferred LP to TL (mean health state utility value: 0.64 vs. 0.56); comparisons of poor outcomes after both treatment approaches revealed that pCRT and TL + aRT were equivalently valued (mean utility value: 0.33 vs. 0.32) [82]. Thus, Hamilton noted that functional treatment outcomes have a greater effect on the utility values assigned to a health state than treatment modality [82]. Therefore, primary reluctance against TL has to be differentiated from definite rejection of TL. This differentiation should be achieved by (1) frank information about the different treatment options and the respective relative outcome probabilities of the individual patient depending on the tumor-related risk factors described and (2) detailed information about the consequences and quality of life after both treatment strategies and the possibilities of rehabilitation. This information should include, among others, an appointment with a speech therapist explaining the different possibilities of voice rehabilitation, an interview with a person who has experienced total laryngectomy and received successful voice rehabilitation, and a discussion with a member of a self-help organization. The decision-making process should be granted the necessary time and should be accompanied by close relatives and/or friends.

A shortcoming of our study is the small number of nonsurgically treated T3 patients (*n* = 18) compared with the number of surgically treated patients (*n* = 103). This limited number of nonsurgically treated patients is because during the time of recruitment of patients, 1998–2004, i.e., twenty years ago, in contrast to neighboring Denmark or Netherlands, in Germany, treatment of laryngeal cancer was mainly surgical. Another shortcoming is that besides OSfL, in surgical and conservative LP patients, only oncological outcomes were reported but not functional outcomes such as tracheostomy and feeding tube dependence. A further weakness of the study is the negative selection bias of older and less healthy patients for pRT. However, in multivariable analysis, age and CCI of patients were included as independent prognostic markers. An advantage of our study is the long follow-up period of up to 17 years and the characteristics of an observational cohort study. Thus, patients were treated according to general medical community practice, which may differ from the highly controlled treatment by skilled investigators in randomized clinical trials. Often, in clinical studies, the results are more favorable and may not be transferred without concern to everyday standard practice [83,84]. Thus, Lefebvre noted that the patients who were included in the VA and RTOG studies were not typical patients with advanced laryngeal cancers who were candidates for TL in daily practice (two-thirds supraglottic tumors, 40% normal vocal cord mobility, only 10% T4, and 80% Karnofsky score >90%) [85].

## 5. Conclusions

Based on data from our study and the literature, outcomes after pRT alone are significantly poorer than those after pCRT or primary surgery in T3 laryngeal cancers. Here, pRT alone might be considered when a T3 patient is not suitable for the addition of chemotherapy (age >70, poor performance status, multimorbidity, especially reduced renal function), when there is strong patient preference and increased willingness for the risk of poor survival outcome, and there is a small T3 (<6 ccm) with no further tumor-related risk factors (vocal cord fixation, cartilage infiltration, N2–3). Otherwise, for T3 laryngeal cancers, the regimen of choice for LP is pCRT. In cases of high-risk criteria, such as tumor volume >12 ccm, vocal cord fixation by direct involvement of the crico-arytenoid joint (Succo pattern III or IV), gross or multiple cartilage infiltration, pretreatment dysphagia/aspiration requiring feeding tube or tracheostomy, or in cases where pCRT is indicated but the patient is not suitable for the addition of chemotherapy, upfront total laryngectomy should be recommended. Further, clinical studies—ideally prospective—validating these selection criteria for LP vs. TL ± a(C)RT are warranted.

## Figures and Tables

**Figure 1 cancers-13-03435-f001:**
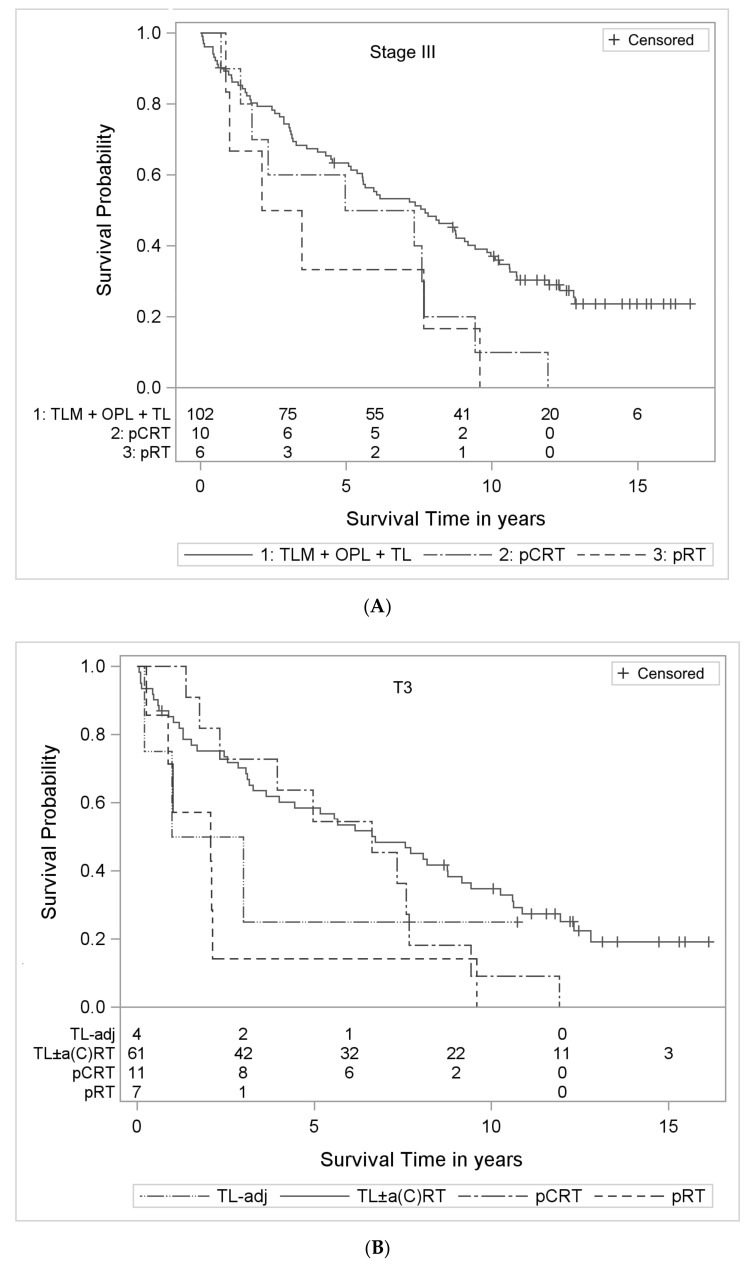
OS after surgical and nonsurgical treatments evaluated over stage III and T3: (**A**) Kaplan–Meier curves with numbers at risk of stage III laryngeal cancer patients in the three treatment arms. TLM: transoral laser microsurgery; OPL: open partial laryngectomy; TL: total laryngectomy; pCRT: primary radiochemotherapy; pRT: primary radiotherapy; (**B**) Kaplan–Meier curves with numbers at risk of T3 laryngeal cancer patients in the three treatment arms. Total laryngectomy (TL) patients were subdivided into TL with stage-appropriate adjuvant treatment (TL ± a(C)RT) and TL without adjuvant treatment despite N2–3 (TL−adj).

**Table 1 cancers-13-03435-t001:** Demographic and clinical characteristics of 757 laryngeal cancer patients.

Characteristic	Category	TLM	OPL	TL	pCRT	pRT	Total
Total		443	59	172	38	45	757
Age (cont) ^a^		62.5 (37–91)	62.1 (34–84)	61.9 (40–83)	61.4 (41–81)	64.9 (40–85)	62.4 (34–91)
Sex	Males	402 (90.7)	57 (96.6)	158 (91.9)	32 (84.2)	36 (80.0)	685 (90.5)
	Females	41 (9.3)	2 (3.4)	14 (8.1)	6 (15.8)	9 (20.0)	72 (9.5)
CCI	0	331 (74.7)	45 (76.3)	114 (66.3)	31 (81.6)	22 (48.9)	543 (71.7)
	1	112 (25.3)	14 (23.7)	58 (33.7)	7 (18.4)	23 (51.1)	214 (28.3)
Localization	Glottic	336 (75.8)	49 (83.1)	49 (28.5)	8 (21.1)	23 (51.1)	465 (61.4)
	Supraglottic	96 (21.7)	7 (11.9)	57 (33.1)	20 (52.6)	14 (31.1)	194 (25.6)
	Subglottic	4 (0.9)	0 (0.0)	8 (4.7)	1 (2.6)	1 (2.2)	14 (1.8)
	Transglottic	4 (0.9)	0 (0.0)	38 (22.1)	6 (15.8)	3 (6.7)	51 (6.7)
	Unknown	3 (0.7)	3 (5.1)	20 (11.6)	3 (7.9)	4 (8.9)	33 (4.4)
T-Stage	1	277 (62.5)	32 (54.2)	5 (2.9)	5 (13.2)	12 (26.7)	331 (43.7)
	2	122 (27.5)	17 (28.8)	34 (19.8)	9 (23.7)	18 (40.0)	200 (26.4)
	3	31 (7.0)	7 (11.9)	65 (37.8)	11 (28.9)	7 (15.6)	121 (16.0)
	4	13 (2.9)	3 (5.1)	68 (39.5)	13 (34.2)	8 (17.8)	105 (13.9)
N-Stage	0	363 (81.9)	54 (91.5)	105 (61.0)	19 (50.0)	30 (66.7)	571 (75.4)
	1	18 (4.1)	0 (0.0)	21 (12.2)	3 (7.9)	4 (8.9)	46 (6.1)
	2	31 (7.0)	2 (3.4)	41 (23.8)	11 (28.9)	8 (17.8)	93 (12.3)
	3	1 (0.2)	0 (0.0)	1 (0.6)	3 (7.9)	2 (4.4)	7 (0.9)
	X	30 (6.8)	3 (5.1)	4 (2.3)	2 (5.3)	1 (2.2)	40 (5.3)
UICC Stage	I	265 (59.8)	31 (52.5)	3 (1.7)	3 (7.9)	10 (22.2)	312 (41.2)
	II	98 (22.1)	17 (28.8)	25 (14.5)	6 (15.8)	15 (33.3)	161 (21.3)
	III	39 (8.8)	6 (10.2)	57 (33.1)	10 (26.3)	6 (13.3)	118 (15.6)
	IV	41 (9.3)	5 (8.5)	87 (50.6)	19 (50.0)	14 (31.1)	166 (21.9)
Adjuvant Treatment	None *	360 (81.3)	52 (88.1)	93 (54.1)	38 (100)	45 (100)	588 (77.7)
	aRT	74 (16.7)	7 (11.9)	62 (36.0)	0 (0.0)	0 (0.0)	143 (18.9)
	aCRT	5 (1.1)	0 (0.0)	17 (9.9)	0 (0.0)	0 (0.0)	22 (2.9)
	aCT	4 (0.9)	0 (0.0)	0 (0.0)	0 (0.0)	0 (0.0)	4 (0.5)

TLM: transoral laser microsurgery; OPL: open partial laryngectomy; TL: total laryngectomy; pCRT: primary chemoradiotherapy; pRT: primary radiotherapy; CCI: Charlson Comorbidity Index; Age (cont) ^a^: age (continuous) in years: mean with minimal and maximal ages; Localization: primary tumor localization; * in some charts the entry was missing, i.e., no adjuvant treatment or unknown status; aRT: adjuvant radiotherapy; aCRT: adjuvant chemoradiotherapy; aCT: adjuvant chemotherapy.

**Table 2 cancers-13-03435-t002:** Rate of adjuvant treatment depending on primary treatment for T3 patients.

Primary Treatment	No Adjuvant Treatment	aRT	aCRT	Total
TL	41 (63.1%)	16 (24.6%)	8 (12.3%)	65 (100%)
OPL	5 (71.4%)	2 (28.6%)	0 (0%)	7 (100%)
TLM	15 (48.4%)	15 (48.4%)	1 (3.2%)	31 (100%)
pCRT	11	-	-	11 (100%)
pRT	7	-	-	7 (100%)

TL: total laryngectomy; OPL: open partial laryngectomy; TLM: transoral laser microsurgery; pCRT: primary chemoradiotherapy; pRT: primary radiotherapy; aRT: adjuvant radiotherapy; aCRT: adjuvant chemoradiotherapy.

**Table 3 cancers-13-03435-t003:** DSS, OS, and OSfL after 5 and 10 years of T3 patients in the different treatment arms.

Treatment Group (*n* *)	5/10-Year DSS (% 95%-CI)	5/10-Year OS (%, 95%-CI)	5/10-Year OSfL (%, 95%-CI)
TLM + OPL + TL (99/103)	76/65 65–83/53–74	61/35 50–69/26–45	n.a.
TL ± a(C)RT (59/61)	75/64 61–84/49–75	58/35 45–70/23–47	n.a.
TL−adj (4)	25/25 1–67/1–67	25/25 1–67/1–67	n.a.
OPL ± a(C)RT (7)	83/83 27–97/27–97	71/36 26–92/5–70	71/36 26–92/5–70
TLM ± a(C)RT (29/31)	84/69 62–94/44–84	67/37 47–81/20–54	49/32 30–66/16–49
pCRT (10/11)	70/26 33–89/1–66	55/9 23–78/1–33	36/9 11–63/1–33
pRT (5/7)	20/20 1–58/1–58	14/14 1–46/1–46	14/14 1–46/1–46

TLM: transoral laser microsurgery; OPL: open partial laryngectomy; TL: total laryngectomy; TL ± a(C)RT: total laryngectomy with stage-appropriate adjuvant chemoradiation or radiotherapy alone; TL−adj: total laryngectomy without adjuvant treatment despite N2–3; pCRT: primary chemoradiotherapy; pRT: primary radiotherapy; 5/10-year DSS: 5- and 10-year disease specific survival, respectively; OS: overall survival; OSfL overall survival with functional larynx. * Differences in numbers of patients between OS/OSfL and DSS due to unknown causes of death.

**Table 4 cancers-13-03435-t004:** (**A**) Univariable Cox regression analysis of OS in stage III; (**B**) Multivariable Cox regression analysis of OS in stage III; (**C**) Univariable Cox regression analysis of OS in T3; (**D**) Multivariable Cox regression analysis of OS in T3.

(A)					
Variable	Category	*p*-Value	HR	95%-CI	
Therapy	pCRT	0.1137	1.710	0.880	3.326
	pRT	0.0031	3.666	1.548	8.682
**(B)**					
**Variable**	**Category**	***p*-Value**	**HR**	**95%-CI**	
Age	10 years units	0.0285	1.325	1.030	1.704
Sex	Female vs. male	0.1795	0.577	0.295	1.288
CCI	1 vs. 0	0.0450	1.671	1.011	2.759
Therapy	pCRT	0.3124	1.443	0.709	2.937
	pRT	0.0006	4.860	1.966	12.013
Localization	supraglottic	0.7406	0.916	0.545	1.539
	subglottic	0.3582	2.014	0.452	8.969
	transglottic	0.5954	1.201	0.610	2.364
	unknown	0.9599	1.023	0.427	2.447
**(C)**					
**Variable**	**Category**	***p*-Value**	**HR**	**95%-CI**	
Therapy	TL−adj	0.6415	1.321	0.409	4.260
	pCRT	0.3304	1.390	0.716	2.698
	pRT	0.0021	3.822	1.626	8.986
**(D)**					
**Variable**	**Category**	***p*-Value**	**HR**	**95%-CI**	
Age	10-year units	0.0476	1.301	1.003	1.688
Sex	Female vs. male	0.7943	0.886	0.358	2.193
CCI	1 vs. 0	0.1604	1.509	0.850	2.680
Therapy	TL−adj	0.8376	1.141	0.323	4.024
	pCRT	0.6329	1.199	0.569	2.529
	pRT	0.0020	4.400	1.716	11.280
Localization	supraglottic	0.8561	0.941	0.490	1.809
	subglottic	0.6227	1.706	0.203	14.323
	transglottic	0.8223	0.913	0.411	2.025
	unknown	0.7149	0.849	0.354	2.039

HR: hazard ratio; 95%-CI: 95%-confidence interval; pCRT: primary chemoradiotherapy; pRT: primary radiotherapy; (**B**) HR: hazard ratio; 95%-CI: 95%-confidence interval; CCI: Charlson Comorbidity Index; pCRT: primary chemoradiotherapy; pRT: primary radiotherapy; references: therapy vs. TLM + OPL + TL. Localization vs. glottic; (**C**) HR: hazard ratio; 95%-CI: 95%-confidence interval; TL−adj: total laryngectomy without adjuvant treatment despite N2–3; pCRT: primary chemoradiotherapy; pRT: primary radiotherapy; (**D**) HR: hazard ratio; 95%-CI: 95%-confidence interval; CCI: Charlson Comorbidity Index; TL−adj: total laryngectomy without adjuvant treatment despite N2–3; pCRT: primary chemoradiotherapy; pRT: primary radiotherapy; references: therapy vs. TL with stage-appropriate adjuvant treatment (TL ± a(C)RT). Localization vs. glottic.

## Data Availability

The datasets generated and analyzed during the current study are available from the corresponding author on reasonable request.

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
