# Peer review of "Chemoradiotherapy but Not Radiotherapy Alone for Larynx Preservation in T3. Considerations from a German Observational Cohort Study"

_cancers, 2021, doi:10.3390/cancers13143435_

Round 1

Reviewer 1 Report

I commend the authors on this new approach to examining optimal survival outcomes in T3 laryngeal cancers. A few comments:

Methods:

  • A statement of ethics approval is needed.

Results & Discussion:

  • I would exclude the TL-adj patients since they refused prescribed treatment. There were only 4 of these patients, so the analysis is really not that meaningful.
  • Do you have results on laryngectomy-free survival for the LP cohort?
  • There appears to be a selection bias for TL vs OPL/TLM. Did you account for that in your survival analysis? Were you able to compare TL+RT to CRT? 
    • This bias needs to be addressed as less aggressive disease is more likely to get TLM vs a TL. 
  • Were neck dissections performed on all surgical patients? Were these standardized?
  • What does of RT was used in the primary and adjuvant setting?

Reviewer 2 Report

Dear members of the editorial board,

I read the manuscript titled “Chemoradiotherapy but not Radiotherapy Alone for Larynx Preservation in T3. Considerations from a German Observational Cohort Study” with interest. It is well written. The use of language is appropriate. The study addresses a relevant clinical topic. There are several and larger/detailed previously published studies about this topic. Still, I would endorse the publication of this paper for its bibliographical value (for a future pooled or meta-analysis), especially for its follow-up time. However, it requires major revision as reasoned in detail below.

Best wishes and kind regards

- Table 1 contains the whole cohort, which is mentioned in the beginning, but later no further statistics were performed with the whole population of these patients. That fills the table with unnecessary information, which hampers the simplicity of the reading experience. I would suggest to focus on the analysed cohort.

- Figures 1a&b: The endpoint OS should be made clear in the heading and/or on the Y-axes of the figures.

- The number/rate of patients who required adjuvant radiotherapy alone and chemoradiotherapy after each primary surgery techniques should be provided.

- The mentioned uni- and multivariate analyses should be provided in structured tabular form.

- It is not logical to use age as a potential prognostic factor for survival analyses (may be okay to use if for other endpoints without the death defined as an event). It reminds the correlation between smaller shoe size and longevity (children are expected to live longer)…

- The functional outcome is not reported, which is crucial in such a paper to be published in 2021. The rate of salvage and functional laryngectomies, tracheostomy and feeding tube dependence (for example at the last follow-up) and most importantly the laryngectomy-free survival should be provided.

- The surgical techniques are provided. What about the radiotherapy techniques (dose/fractionation, 2D, 3D, IMRT, total treatment time respected? etc.) and chemotherapy regimens (cisplatin weekly, three-weekly, any patients receiving induction chemotherapy etc.)? It is also interesting that there is no radiation and/or medical oncologist among the co-authors. Did the patients receive chemo- and/or radiotherapy in centers with high-case volume or were they scattered among different private/smaller clinics without dedicated head & neck radiation or medical oncologists? This is known to directly influence tumor control and survival.

- I sincerely congratulate the authors for their enthusiasm and care for detail. But the discussion section is too long for such a modest retrospective cohort study. There are lots of recently published review papers addressing these issues in depth. It is nice to see the important subheadings covered, but this chapter can be significantly shortened by citing these papers for recommended in-depth reading. Moreover, in its current form, the paper also covers topics in detail which are not addressed by the study itself (for example the cartilage invasion, tumor volume etc.).

- Throughout the manuscript, especially in the Discussion section, (primary) “surgery” is directly or indirectly mentioned to be a possible superior treatment over non-surgical therapy. This leads the reader to intuitively imagine that these patients are operated, and the problem is solved. On the contrary, categorically speaking, surgery alone is an inferior therapy if not complemented with appropriate adjuvant treatment. A significant number of these patients end up getting unnecessary trimodal treatments and subsequently increased lifelong morbidity, whereas they could have been treated with less intensive bimodal non-surgical treatment. On the other side of the coin, some patients have to undergo salvage surgery after a failed primary chemo-radiotherapy, which is sometimes not considered as a possibility in the beginning. These two issues cause their corresponding problems in the practice, respectively: A) Patients are informed about primary surgery by downplaying the probability of the necessity of an adjuvant therapy (and what it means in terms of acute and especially late toxicity), or not mentioned at all (Then comes the surprise at the post-op/post-pathology discussion). B) At the initial discussion with the patients, salvage surgery is presented as a back-up solution to the offered chemoradiotherapy “in case it doesn’t work (Don’t worry)”. If we are talking about shared decision-making with the patients, we as surgical, medical and radiation oncologists have to internalize these facts and nuances in order to correctly inform the patients. Actually, there is a nice section under the Discussion section about informed and shared decision making. But maybe, the authors would like to carefully reformulate the use of these terms throughout the manuscript as well.

- A manuscript about this subject is expected to at least briefly mention the editorial by Licitra et al. and their criticism on the RTOG (Forastiere) trial.

- The authors advocate up-front total laryngectomy for elderly and co-morbid patients. What about their compliance for post-TL rehabilitation? Moreover, several times, the decreased value of concomitant chemotherapy with primary RT was shown to be due to competing risk factors for death, rather than a biological loss of effectiveness.

Round 2

Reviewer 2 Report

Dear members of the editorial board,

I thank the authors for their amendment. All questions and comments are addressed adequately. As a minor methodological issue, I still do not agree with the use of age as a parameter in the Cox proportional models for survival endpoints, but I understand the counterargument brought by the authors. Without delving into an unnecessary philosophical discussion about statistical methodology, I would gladly endorse the publication of this manuscript.

Best wishes and kind regards